# Uses of FT-MIR Spectroscopy and Multivariate Analysis in Quality Control of Coffee, Cocoa, and Commercially Important Spices

**DOI:** 10.3390/foods11040579

**Published:** 2022-02-17

**Authors:** Lucero Azusena Castillejos-Mijangos, Aracely Acosta-Caudillo, Tzayhrí Gallardo-Velázquez, Guillermo Osorio-Revilla, Cristian Jiménez-Martínez

**Affiliations:** 1Escuela Nacional de Ciencias Biológicas, Instituto Politécnico Nacional, Av. Wilfrido Massieu Esq. Cda. Manuel Stampa s/n, Alcaldía Gustavo A. Madero, Ciudad de Mexico C.P. 07738, Mexico; castillejos767@gmail.com (L.A.C.-M.); aracibq.508@hotmail.com (A.A.-C.); osorgi@gmail.com (G.O.-R.); 2Departamento de Biofísica, Escuela Nacional de Ciencias Biológicas, Instituto Politécnico Nacional, Prolongación de Carpio y Plan de Ayala s/n, Col. Santo Tomás, Ciudad de Mexico C.P. 11340, Mexico

**Keywords:** spices, coffee, cocoa, quality control, adulteration, FT-MIR, multivariate analysis

## Abstract

Nowadays, coffee, cocoa, and spices have broad applications in the food and pharmaceutical industries due to their organoleptic and nutraceutical properties, which have turned them into products of great commercial demand. Consequently, these products are susceptible to fraud and adulteration, especially those sold at high prices, such as saffron, vanilla, and turmeric. This situation represents a major problem for industries and consumers’ health. Implementing analytical techniques, i.e., Fourier transform mid-infrared (FT-MIR) spectroscopy coupled with multivariate analysis, can ensure the authenticity and quality of these products since these provide unique information on food matrices. The present review addresses FT-MIR spectroscopy and multivariate analysis application on coffee, cocoa, and spices authentication and quality control, revealing their potential use and elucidating areas of opportunity for future research.

## 1. Introduction

Spices in ancient Egypt and Mesopotamia were used as medicine, currency, and food preservatives. Later were used for seasoning, aromatizing, and coloring different dishes and beverages [1,2]. According to the U.S. Food and Drug Administration [3], spices are aromatic vegetable substances in the whole, crushed, or ground form that flavor food without adding nutritional value. These are true to name, and from them, no portion of any volatile oil or other flavoring principle is removed. Examples of plant parts used as spices are bark (cinnamon), buds (cloves), flowers (saffron), fruits (bell pepper, chili), pods (vanilla), rhizomes (ginger, turmeric), seeds (pepper, cardamom), and bulbs (onion, garlic) [4,5,6].

In addition to their organoleptic properties (flavor, aroma, color, and pungency), spices provide health benefits to consumers due to their antioxidant, antimicrobial, antidiabetic, antimutagenic, anti-inflammatory, and immunomodulatory effects [7,8,9,10,11]. Moreover, contrary to what was believed in ancient times, research demonstrated that spices are a rich source of proteins, lipids, vitamins, and minerals [12].

Organoleptic and nutraceutical properties of spices are mainly attributed to their bioactive compounds such as crocin, eugenol, piperine, curcumin, cinnamaldehyde, capsaicin, quercetin, kaempferol, vanillic acid, caffeic acid, coumaric acid, among others [5,11,13]. As a result, they have salient applications in the food and beverage, cosmetics, perfumery, pharmaceutical, and medical industries [2,14].

Spices are a highly demanded product worldwide, whose market is valued at approximately 12,700 million dollars with a favorable projected growth for subsequent years [15,16]. The high economic value of spices makes them susceptible to fraud and adulteration at different processing stages (production, storage, transport, and distribution) [17]. Adulteration is the act of intentionally adding undeclared compounds or substances for major economic gain at the expense of food quality and consumers’ health. Common adulterated spices are saffron, vanilla, turmeric, cardamom, and paprika [1,18]. Usually, spices are adulterated with plant compounds similar in appearance but with a lower value (e.g., peels, leaves, petals, seeds, and grains), old batches of the same spice, inferior varieties, and other products (e.g., ground stones, brick dust, sawdust, flour, and starch); in addition to artificial colors (Sudan I–IV, Orange II, Metanil Yellow, Basic Red 46, Rhodamine B, and Malachite Green), which can be carcinogenic and genotoxic [13,16,18,19,20]. On the other hand, although they are not considered spices, cocoa and coffee have an intense aroma and flavor and are highly consumed around the world. Due to the rise in demand in recent years, it is important to ensure that the quality of the products is maintained since these are often adulterated with products with similar physical characteristics after being roasted and ground.

Besides authentication (absence of adulteration), quality control of coffee, cocoa, and spices includes:Evaluation of sensory and physical characteristics, i.e., solubility, density, presence or absence of external materials;Analysis of physicochemical parameters such as moisture content, ash, protein, and fat;Presence and concentration of bioactive compounds;Microbiological analysis and absence of natural contaminants such as mycotoxins and heavy metals;Classification of spices based on their geographical origin and variety [19,21].

Ensuring the quality of coffee, cocoa, and spices requires the use of appropriate technologies. Conventional methods for authentication and quality control range from simple techniques such as Kjeldahl, Soxhlet, and UV/vis to more specialized ones such as high-performance liquid chromatography (HPLC), gas chromatography (GC), capillary electrophoresis (CE), mass spectrometry (MS), X-ray fluorescence (XRF), inductively coupled plasma (ICP), proton transfer reaction (PTR), matrix-assisted laser desorption/ionization-time-of-flight (MALDI-TOF), and DNA analysis, among others. Nevertheless, all these require prior sample preparation and detailed knowledge; besides, the procedure is slow, destructive, expensive, pollutant, and hazardous for the analyst [12,20,22]. Consequently, better alternatives should be implemented, namely infrared spectroscopy (IR) [23,24,25].

Mid-infrared (MIR) and near-infrared (NIR) spectroscopy are widely used in food quality control because spectra analysis in the fingerprint region provides unique information to differentiate compounds, even if two molecules have the same functional groups [26,27,28,29]. There is a greater amount of research on the application of NIR spectroscopy to different spices compared to MIR spectroscopy [30,31,32,33,34,35,36,37,38]. Nonetheless, studies on MIR spectroscopy have increased over the years since this method can identify trace elements. Additionally, MIR spectra are simpler to interpret and present greater specificity and selectivity than NIR [27,39,40]. Therefore, the present work assesses the application of Fourier transform mid-infrared (FT-MIR) spectroscopy on the most economically important spices.

We conducted a review of the literature on FT-MIR spectroscopy combined with multivariate analysis methods for authentication and quality control of coffee, cocoa, saffron, turmeric, vanilla, and other spices. In addition, to the advantages and limitations of using this technology.

## 2. Mid-Infrared (MIR) Spectroscopy

MIR spectroscopy is a versatile technique based on the vibration of molecular bonds of compounds that covers the 4000–400 cm^−1^ region of infrared (IR) spectrum. Vibrational motions are classified into two groups: stretching (symmetric and asymmetric) and bending (scissoring, rocking, wagging, and twisting) [41]. Stretching vibration occurs with the change in bond length, while bending vibrations generate a change in angle [2]. The absorption bands of the MIR spectrum are related to those vibrations, and their intensity provides information on the concentration of a compound in a food matrix; hence, MIR spectroscopy can be applied for qualitative and quantitative analysis [28]. Moreover, the MIR region is robust and reproducible, which allows identifying with certainty minimal differences in the chemical composition of a sample [42]. The MIR spectrum is divided into four major regions: stretching (4000–2500 cm^−1^), triple bonds (2500–2000 cm^−1^), double bonds (2000–1500 cm^−1^), and fingerprint (1500–400 cm^−1^). In the fingerprint region, unique absorbance patterns of each sample are presented, allowing the distinction of similar substances [27,43].

Advances in MIR spectroscopy led to the incorporation of FT-MIR spectrometer in food. FT-MIR is a practical, sensitive, and fast equipment that has a high signal-to-noise ratio (SNR) and provides well-defined spectra in milliseconds without losing resolution [43,44]. FT-MIR spectroscopy combined with multivariate analysis enables the rapid characterization and classification of foods, detecting compounds present at concentrations of parts per billion [42,45].

FT-MIR spectroscopy allows the analysis of solid, liquid, or gaseous samples using different sampling techniques (transmission or reflectance) that improve the resolution of the obtained spectra [44,46]. Transmission sampling techniques require sample preparation, including NaCl, KBr, CaF_2_, CsBr, and ZnSe cells for liquids; KBr pellets and Nujol mulls for solids; and glass cells with NaCl/KBr-coated windows for gases [42,43,47,48]. On the other hand, reflectance sampling comprises diffuse reflectance (DRIFT), specular reflectance, and attenuated total reflectance (ATR) [49]. ATR is the most widely used sampling technique in FT-MIR spectroscopy because it is easy to use, non-destructive, and fast; furthermore, samples require minimal or no preparation, which improves reproducibility by reducing spectral variation. MIR spectra of solid foods, liquids, powders, pastes, and viscous fluids can be obtained through ATR-FT-MIR spectroscopy [45,50] using diverse crystal materials (typically made of ZnSe, ZnS, Ge, Si, or diamond) according to sample type [27].

## 3. Multivariate Analysis

Combining instrumental techniques with multivariate analysis can extract qualitative and quantitative information from MIR spectra, which enhances the development of composition, adulteration, and traceability methods for food analysis [51,52,53].

Multivariate analysis uses mathematical and statistical tools to provide complete information from large chemical data sets [54,55]. First, it is necessary to interpret the MIR spectra to identify the compounds from the sample and choose the best processing strategies for the multivariate analysis model [56,57]. Subsequently, MIR spectra are pretreated to eliminate or reduce variations produced by the nature of the sample or environmental conditions during the measurement. Spectral pretreatments include baseline correction, normalization, smoothing, and derivative [58,59].
-Normalization: Reduces spectral variation generated by sample manipulation. Multivariate analysis programs offer different normalization methods, with multiple scatter correction (MSC) and standard normal variable transformation (SNV) being the most used. MSC separates the light scattering signals from the chemical absorption signals of the sample, making the chemometric analysis insensitive to changes in the baseline spectrum. At the same time, SNV removes multiplicative scattering and particle size interferences. Both normalizations lead to similar results, so it is unnecessary to test both when optimizing a model [60,61];-Baseline correction: Removes baseline slopes from spectra commonly generated by IR beam scattering. Baseline correction attempts to correct spectra without distorting band intensities or introducing discontinuities [62];-Smoothing: Removes or reduces noise from MIR spectra using algorithms but also degrades the resolution of the spectrum by broadening its characteristics. The amount of noise in the spectrum is related to the number of scans performed, the higher the number, the lower the analytical signal’s noise. The most commonly used algorithm is the Savitzky–Golay algorithm, which aims to minimize the distortion of the spectrum; the degree of smoothing to be applied will depend on the bandwidth, care must be taken since a very high smoothing excessively broadens the bands generating a loss of resolution, so its use is not recommended in very sharp bands [53,60];-Derivative: It can be first or second order and is used to correct for changes in the baseline and reduce the effect of overlapping bands. Derived spectra present sharper features compared to the original spectra. The first and second-order derivatives eliminate any deviation in the baseline. However, the second-order derivative is also helpful in identifying band positions in complex regions [53,63].

After applying the necessary spectral pretreatments, chemometric models are developed using classification tools such as principal component analysis (PCA) and hierarchical cluster analysis (HCA) and quantification tools such as principal component regression (PCR), multiple linear regression (MLR), and partial least squares (PLS) regression [29,42,64]. The most used chemometric tools in food for classification (PCA) and quantification (PLS) analysis were briefly addressed [52,65].

### 3.1. Principal Component Analysis (PCA)

Principal component analysis (PCA) is a traditional chemometric technique that improves the understanding of an extensive spectral data set by reducing it to a small number of new uncorrelated variables (orthogonal variables) called principal components (PCs), which contain the maximum information (maximum variance) of the original data set [66,67]. The first PC explains most of the variance; the second PC explains the information not modeled by the first PC until reaching the last PC, which mainly describes the noise [56,65,68]. Each sample is projected onto a graphical representation whose value of its coordinates is called scores or loadings. The graphical display of scores helps find clusters in a data set and identify outliers or misclassified samples that can be reanalyzed or removed from the data set to be processed [57,69]. In addition to PCA, there are other frequently used tools for sample classification or discrimination such as linear discriminant analysis (LDA), hierarchical cluster analysis (HCA), factor discriminant analysis (FDA), canonical variant analysis (CVA), artificial neural networks (ANN), partial least squares discriminant analysis (PLS-DA), random forest (RF), quadratic discriminant analysis (QDA), k-nearest neighbors (k-NN), supervised local linear embedding (SLLE), and support vector machines (SVM) [28,69,70].

### 3.2. Partial Least Squares Regression (PLS)

Partial least squares (PLS) regression is the most popular multivariate calibration tool for data processing and quantitative model development [57,58,64]. From the spectral data, the PLS algorithm calculates new uncorrelated latent variables (LV) called factors, which explain the maximum covariance between spectral data and analytical data obtained from a reference method. The first LV maximizes the covariance between spectral and analytical data while the latter explains a decreasing residual variance [56,58,69,70]. When evaluating a PLS regression model, the standard deviation of calibration error (SDEC) calculated from the samples with which the model was built and the standard deviation of prediction (SDEP) calculated with samples different from the one used in the construction of the model are considered [57]. Other algorithms used less frequently for the development of quantitative models are principal component regression (PCR), multiple linear regression (MLR), and support vector machine (SVM) regression [28,70].

Ensuring the prediction accuracy of unknown samples requires a multi-step process: selecting a set of representative FT-MIR spectra, determining the property of interest using a reference method, choosing the calibration (3/4) and validation samples (1/4), performing the data pretreatment, applying the classification or quantification algorithm, and validating the model [58,71].

## 4. FT-MIR Spectroscopy Applications in Coffee, Cocoa, and High-Priced Spices

Globally, the highest-priced spices are obtained from not widely cultivated plants or demand meticulous and laborious production processes, e.g., saffron, vanilla, turmeric, paprika, cinnamon, and cardamom [16,17]. The following sections address FT-MIR spectroscopy and applications to determine the authenticity and quality of those spices. Additional studies on coffee and cocoa are included for their high economic impact and broad use [72].

### 4.1. Coffee

Coffee is one of the main traded crops and the second most consumed beverage worldwide [73,74]. The coffee plant belongs to the genus *Coffea* L., which comprises more than 100 species, but only *Coffea arabica* (Arabica coffee) and *Coffea canephora* (Robusta coffee) are of commercial and economic importance [72,75]. After harvesting, coffee cherries undergo processing (dry, semi-wet, or wet) to separate the coffee beans (green or raw) from the fruit and reduce their moisture content (up to 10–12%). Likewise, it facilitates coffee beans transportation without a loss of quality [76,77]. In the international market, the leading coffee exporting countries are Brazil, Vietnam, Indonesia, Colombia, and Ethiopia [75,78].

Green coffee beans require a roasting process for consumption. Coffee roasting softens the bean, facilitates the detachment of the husk, and augments coffee’s characteristic color, flavor, and aroma. During this method, some coffee compounds are degraded (proteins, trigonelline, chlorogenic acids, and phenols), and others are formed (melanoidins and volatile compounds) [77,78,79]. Initially, roasting is carried out at 180 °C aiming to reduce the moisture content of the beans to 1.5–5%; then, the temperature is raised to 200–300 °C commencing Maillard and caramelization reactions, which lead to the formation of volatile substances that generate coffee characteristics. Once the desired roasting is obtained, coffee beans are cooled by water or air to stop the biochemical reactions and avoid further degradation. Roasted coffee beans consist of polysaccharides (38–42%), lipids (11–17%), proteins (7.5–10%), aliphatic acids (1.6%), chlorogenic acids (2.5–3.8%), caffeine (1.3–2.4%), trigonelline (0.7–1%), minerals (4.5–4.7%), volatile compounds (0.1%) and melanoidins (23–25%) [80,81]. Besides providing health benefits, (e.g., vasoconstrictor, neuroprotective and neurostimulator, antioxidant, anti-inflammatory, and anticarcinogenic), caffeine, trigonelline, chlorogenic acids, and volatile compounds content are related to coffee quality [79,81,82,83,84]. Approximately 60–80% of the world’s coffee production is obtained from *Coffea arabica* species and the other from *Coffea canephora* [77,85]. Both commercial species of coffee differ in composition. *C. canephora* has a high content of caffeine and chlorogenic acids; therefore, *C. arabica* quality is considered superior and double the price of *C. canephora* (Robusta coffee). Moreover, the geographical origin of *C. arabica* can further raise its cost [86,87,88]. After roasting and grinding, those differences become imperceptible; as a result, *C. arabica* beans can be adulterated with *C. canephora* to reduce production costs [73]. Other low-grade products might be added to *C. arabica*, for example, coffee husks and grounds, corn wheat, barley, soybean, and rye [74,86,89]. Coffee quality and authenticity are generally measured through chromatography (HPLC and GC), spectroscopy (UV-vis, NMR, fluorescence, IR, and Raman), inductively coupled plasma (ICP), real-time PCR, atomic absorption spectrometry (AAS), and mass spectrometry (MS) [84,87,88].

FT-MIR spectroscopy and multivariate analysis are applied for coffee authentication and quality control (Table 1). Wang et al. [90] implemented these techniques to detect and quantify adulteration of Kona coffee with lower-quality coffee. They used ground and brewed coffee in spectra range (1900–800 cm^−1^), first and second derivative pretreatments, and PCR and PLS analysis to calibrate the models. The best calibration results were obtained with the spectra of brewed coffee (R^2^ = 0.999), applying the second derivative and PLS algorithm. The unfavorable results for ground coffee spectra were explained by low spectral data precision due to particle shape and size.

Ribeiro et al. [91] performed discrimination on commercial coffee samples according to their caffeine content (caffeinated and decaffeinated) and classification based on roasting degree. They used DRIFT, smoothing, and multiplicative scatter correction (MSC) pretreatments and selected the spectral regions 3600–2820 and 1800–784 cm^−1^. Later, they effectuated the discrimination model using PCA and the classification model with PLS-DA achieving 100% classification of the external validation samples. Craig et al. [92,93] developed multivariate analysis models to discriminate defective and non-defective green coffee beans by comparing different sampling techniques (KBr pellets, ZnSe ATR, and DRIFT) and multivariate analyses (PCA, LDA, and HCA). They executed baseline correction and normalization in all models, except in the KBr pellets’ model, where the first derivative was applied. Subsequently, they developed the classification models based on PCA and LDA algorithms using the DRIFT data, whose predictive capacity ranged between 95% and 100% [94]. Finally, the authors classified Arabica coffee according to cup quality. They used ZnSe ATR sampling, baseline correction, normalization, MSC, first and second derivative; employed PCA to separate Arabica and Robusta coffee; and built the classification models with PLS-DA, which presented 100% sensitivity and specificity in calibration but 67–100% in validation [95]. Reis et al. [96,97,98] conducted diverse studies on roasted coffee, adulterants (coffee husk, coffee grounds, roasted corn, and roasted barley), and coffee-adulterant mixtures using 3200–700 cm^−1^ spectral region and DRIFT sampling. Through PCA, they separated the pure samples and each of the adulterants and identified the spectral regions with significant variations. Simultaneously, they elaborated LDA classification models for roasted coffee, pure adulterant, and coffee-adulterant mixtures, obtaining 100% prediction and recognition capacity. After, they tested a PLS algorithm to predict the level of adulteration of the samples (1–66%, *w*/*w*), achieving high calibration (0.99) and validation (0.98) coefficients and low percentage of error (1.23% for calibration and 2.67% for validation). Researchers simultaneously quantified, with the ZnSe ATR sampling technique, the above-mentioned adulterants in roasted coffee, obtaining correlation coefficients of 0.99 for validation and calibration and a percentage of error of 0.69% for calibration and 2.00% for validation [99]. Lastly, they performed a discriminant analysis of the data by comparing DRIFT and ZnSe ATR techniques, employing hierarchical models (HM), and then spectral data fusion (DF). The percentage of misclassified samples decreased to 0% after DF [100].

Brondi et al. [101] introduced two similar models to detect and quantify adulteration of roasted coffee with corn (0.5–40% *w*/*w*). The models were built from data collected with differential scanning calorimetry (DSC) and FT-MIR techniques coupled to PCA and PLS. They were able to discriminate between pure coffee and coffee-corn mixtures with both techniques. The FT-MIR model presented lower cross-validation mean square error (RMSECV = 2.7%) than DSC; therefore, it has a great application potential. Another approach proposed by Link et al. [102] classified Arabica coffee samples based on geographic and genotypic origin using KBr pellets sampling. Spectra (1900–800 cm^−1^ region) were pretreated with normalization, baseline correction, and smoothing. They employed a radial basis function (RBF) network, an artificial neural network (ANN), to build the classification models, getting better geographic (100%) and genotypic (94.44%) classification results compared with soft independent modeling of class analogy (SIMCA) and multilayer perceptron (MLP). Bona et al. [103] also performed geographical classification of Arabica coffee through SVM and NIR and MIR spectroscopy. The best results were obtained with the NIR-SVM approach, where all samples were correctly validated.

Correia et al. [104] quantified Robusta coffee content in adulterated Arabica coffee samples using ZnSe ATR sampling. They analyzed the samples with FT-MIR and developed a multivariate PLS model that presented low detection (LOD = 1.29%) and quantification (LOQ = 4.3%) limits and a high coefficient of determination (R^2^ = 0.9635) in the cross-validation. They further evaluated the samples by negative-ion mode electrospray ionization Fourier transform ion cyclotron resonance mass spectrometry (ESI(-)FT-ICR MS) and modeled the data by univariate analysis, obtaining slightly better results. However, FT-MIR is a simpler technique than ESI(-)FT-ICR MS.

Medina et al. [105] compared three spectroscopic techniques (1H-NMR, NIR, and MIR) for coffee discrimination according to species (Arabica and Robusta) and origin (Colombia and other countries). FT-MIR spectra were collected in the 1800–800 cm^−1^ range and pretreated with normalization and second derivative; the models were built using PCA and PLS-DA. All techniques successfully discriminated the samples by species, although discrimination by origin FT-MIR and 1H-nuclear magnetic resonance (1H-NMR) showed better results than NIR. The authors emphasized that FT-MIR is a faster and cheaper technique in comparison with 1H-NMR.

Obeidat et al. [106] performed another model for origin discrimination in green coffee samples from different countries (Brazil, Colombia, Ethiopia, Kenya, and Yemen) applying ZnSe ATR. They used the 4000–600 cm^−1^ spectral region pretreated with normalization. PCA algorithm allowed to identify the bands with greater spectral variation (2920–2850 and 1745 cm^−1^), enabling successful discrimination of samples based on origin.

Belchior et al. [107], with the assistance of a panel of tasters, built a model of espresso coffee discrimination according to its sensory characteristics. Unlike other researchers, they used commercial capsules of different coffee brands and diverse roasting degrees. Tasters evaluated the sensory quality of espresso coffee on a 5-point scale. The spectra of the beverages were collected in the 3000–600 cm^−1^ region, measured with ZnSe ATR. The data were pretreated with autoscaling and analyzed by PCA to group the samples according to aroma and flavor. Then, they developed discrimination models for each sensory attribute with PLS-DA, which showed high sensitivity and specificity for calibration and validation and low classification errors in cross-validation. The same authors built a model for predicting quality scores in beverages prepared from specialty samples of green Arabica coffee (quality score of 81–91 points established by cuppers). FT-MIR spectra were obtained under the same conditions as their previous work. PLS algorithm results were satisfactory for both calibration (R^2^ = 0.99 and RMSEC = 0.23) and validation (R^2^ = 0.97 and RMSEP = 0.23) [108]. Lastly, Flores-Valdez et al. [109] identified and quantified adulterated samples of Arabica coffee with corn, barley, soybeans, oats, rice, and coffee husks at levels of 1–30%. MIR spectra were obtained with diamond ATR and processed with atmospheric filter pretreatments, normalization, smoothing, and baseline correction; using the spectral regions of 3500–2800 and 1800−800 cm^−1^, they optimized a discrimination model applying a SIMCA algorithm and developed PCR, PLS1, and PLS2 algorithms for quantification models. The discrimination model presented an accuracy of 100% and differentiated all adulterants, while the best quantification model was obtained with the PLS1 algorithm with outstanding calibration (R^2^ = 0.99 and SEC = 0.39–0.82) and validation results (R^2^ = 0.99 and SEP = 0.45–0.94).

**Table 1 foods-11-00579-t001:** Applications of FT-MIR spectroscopy in coffee quality control.

Spectral Range (cm^−1^)	Sampling Technique	Algorithm	Purpose of the Analysis	Reference
Arabica coffee variety Kona typica				
1900–800	ZnSe ATR	PCR PLS	Detection and quantification of adulteration of coffee grown in Kona, Hawaii, with coffee from another region.	[90]
Brazilian coffee				
3600–2820 1800–784 1900–800	DRIFT	PCA PLS-DA	Discrimination of decaffeinated coffee and classification according to roasting degree.	[91]
	KBr pellets	RBF (ANN)	Coffee classification by geographic and genotypic origin.	[102]
Green Arabica coffee				
4000–700	ZnSe ATR DRIFT	PCA, LDA HCA	Discrimination of immature coffee (defective) and mature coffee (non-defective).	[92]
4000–700	KBr pellets ZnSe ATR, DRIFT	PCA HCA	Discrimination of defective and non-defective coffee using three different sampling techniques.	[93]
3600–600	DRIFT	PCA, LDA	Discrimination of defective and non-defective roasted coffee.	[94]
1800–800	KBr pellets	SVM	Geographical classification of different coffee genotypes.	[103]
4000-600 2920–2850 1745	ZnSe ATR	PCA	Discrimination of coffee beans according to their origin (Brazil, Colombia, Ethiopia, Kenya, and Yemen).	[106]
3000–900	ZnSe ATR	PLS	Prediction of quality scores given by cuppers for coffee beverage samples.	[108]
Roasted Arabica coffee				
3200–700	DRIFT	PCA LDA	Discrimination between roasted coffee, corn, coffee husk, coffee-corn, and coffee-husk blends.	[96]
3200–700	DRIFT	PCA LDA	Discrimination between roasted coffee, coffee husks, coffee grounds, corn, barley, and coffee-adulterant blends.	[97]
3200–700	DRIFT	PLS	Prediction of adulteration levels of roasted coffee with different adulterants (pure and blended).	[98]
4000–700	ZnSe ATR	PLS	Simultaneous quantification of four adulterants (coffee husk, coffee grounds, barley, and corn) in roasted coffee.	[99]
4000–525	Diamond ATR	PCA PLS	Detection and quantification of adulteration of roasted coffee with corn.	[101]
3200–700 4000–600	DRIFT ZnSe ATR	PLS-DA HM DF	Discrimination between roasted coffee and adulterated coffee using two sampling techniques and merging data.	[100]
	ZnSe ATR	PCA PLS-DA	Classification of cup quality of coffee with different roasting degrees.	[95]
3500–2800 1800–800	Diamond ATR	SIMCA PCR PLS1, PLS2	Identification and quantification of adulterated coffee with coffee husks, corn, barley, soybeans, oats, and rice.	[109]
Arabica and Robusta coffee				
4000–600 2000–1500 3000–2750	ZnSe ATR	PLS	Quantification of Robusta coffee content in blends with Arabica coffee.	[104]
1800–800	ATR	PCA PLS-DA	Comparison of three spectroscopic techniques (1H-NMR, NIR, and MIR) for the discrimination of coffee by species and origin.	[105]
Commercial coffee capsules				
3000–600	ZnSe ATR	PCA PLS-DA	Discrimination of espresso coffee according to sensory characteristics.	[107]

PCR: principal component regression; PLS: partial least squares regression; PCA: principal component analysis; PLS-DA: partial least squares discriminant analysis; LDA: linear discriminant analysis; HCA: hierarchical cluster analysis; ANN: artificial neural networks; HM: hierarchical models; DF: data fusion; SVM: support vector machine; SIMCA: soft independent modeling of class analogy; PLS1: partial least squares with single y-variables; PLS2: partial least squares with multiple y-variables.

### 4.2. Cocoa

Cocoa beans are used to produce chocolate; these are native to South America but domesticated in Central America. Afterward, the Spanish sped it to Europe and then distributed it to other countries. Nowadays, it is mainly grown in the hot and humid regions of Africa [110], Central and South America, and Asia [111]. There are three main varieties of cocoa: Criollo, Trinitario, and Forastero; the Criollo variety is not produced as much as other varieties despite the suitable quality of its cocoa; this is mainly grown in America. On the other hand, Forastero is produced mainly in Africa; even if its cocoa is not as suitable as Criollo, it has better yields. Lately, Trinitario variety is a cross between Criollo and Forastero, and it yields cocoa of reasonably suitable quality. Although the quality of cocoa depends on the variety of origin, another determining factor is the suitable harvest and proper drying and fermentation. During fermentation, the conditions are provided for the characteristic flavor and aroma of chocolate to develop in the grain through the microorganisms that intervene in the process. A correct process could produce suitable cocoa quality [112].

Due to the increase in the demand for cocoa, it is important to have a technique that allows determining the suitable quality of the product or a correct classification of the kind of variety. Table 2 summarizes the works on FT-MIR spectroscopy application to cocoa and chocolate.

Batista et al. [115] used cocoa beans spontaneously fermented and inoculated with *Saccharomyces cerevisiae* to quantify the antioxidant capacity and the total phenolic compounds of the beans as well as the chocolate produced from them. Results indicated variations in phenolic composition between spontaneously fermented and inoculated samples. The PLS model for total phenolics and antioxidant capacity prediction showed a correlation coefficient >0.94.

Santos et al. [113] performed models for predicting cocoa solids in chocolate, which showed excellent prediction and generalization capability for commercial samples by applying PLS to the MIR data set and reported that the cocoa solids content in 14% of tested chocolates differed in more and 10% of the content presented on the label. On the other hand, Hu et al. [114] built five PLSR models and cross-validated them to quantify catechin, antioxidant capacity, and total phenolics in chocolate, achieving suitable prediction capability for DPPH (R^2^_p_ = 0.89), ORAC (R^2^_p_ = 0.90), Folin–Ciocalteu (R^2^_p_ = 0.88) and (+)-catechin (R^2^_p_ = 0.86) but low accuracy in prediction of (−)-epicatechin (R^2^_p_ = 0.72). Finally, Mandrile et al. [116] used NIR, FT-MIR, and inductively coupled plasma-optical emission spectroscopy (ICP-OES) in conjunction with PCA to authenticate the geographical origin of cocoa shells through their molecular and elemental composition. The best classification results were obtained using the three spectroscopic techniques and PLS-DA to merge the PCA data obtained with each technique and were for Central African samples with an accuracy of 0.84.

### 4.3. Saffron

Saffron is derived from the dried red stigmas of the *Crocus sativus* flower [117]. Saffron flowers do not naturally reproduce and require human handling to subsist, while harvesting stigmas is generally performed manually, making it a delicate process [118,119]. Nonetheless, their ability to add flavor, aroma, and color to foods makes them a largely demanded spice [120]. Saffron, also called “red gold,” is the most expensive spice worldwide (valued at USD 40–60/g) yet highly monopolized by Iran with 90%, followed by India, Afghanistan, Greece, Morocco, Spain, and Italy [117,121,122].

In addition, its antioxidant, anti-inflammatory, and antigenotoxic effects attract the attention of medical and pharmaceutical industries [117,123], having a beneficial influence on cardiovascular and respiratory diseases, metabolic syndrome, depression, anxiety, premenstrual syndrome, digestive disorders, and different types of cancer [124,125,126]. Despite the multiple benefits of saffron, it should be consumed moderately as high doses may cause toxicity [124]. Four bioactive compounds provide saffron sensory and nutraceutical properties: crocin, a carotenoid with high antioxidant activity responsible for the red coloring of the stigmas (or orange when dissolved in water and golden-yellow when mixed with food); crocetin, a carotenoid dicarboxylic acid precursor of crocin; safranal, the main metabolite of saffron, which has a pungent aromatic note and is a potent antioxidant and an anticarcinogen; and picrocrocin that provides a bitter taste and whose hydrolysis during drying allows the release of safranal [125,126,127,128]. The detection and quantification of these bioactive compounds in conjunction with the geographical origin determine the quality of saffron, classified into categories I (high quality), II (medium quality), and III (lower quality) [123,124,126]. However, saffron may be adulterated by adding floral parts of the same (styles and stamens) or other species (calendula, safflower, buddleja, madder, and gardenia), cheaper spices (turmeric and paprika), corn, beet fibers, or artificial colorants that may be toxic to human health, such as Allura Red, Azorubine, Erythrosine, Amaranth, Carminic acid, Tartrazine, Sunset Yellow, Sudan I–IV, Ponceau 4R. The addition of inorganic salts and immersion of saffron in glycerin or syrups to increase weight is also common [120,121,124,129].

Authentication and quality are certified in the international market according to ISO 3632-1/2 [120,123,126]. Saffron quality control can be evaluated through physical (morphological inspection, sensory evaluation, colorimetry, and gravimetry), chromatographic (TLC, HPLC, and GC), molecular (real-time PCR, LAMP, and RAPD-SCAR), sensor-based (E-nose and E-tongue) and spectroscopic (UV-vis, NMR, Raman, MS, FT-NIR, and FT-MIR) methods [22,124]. Table 3 summarizes FT-MIR spectroscopy use for saffron quality control.

Anastasaki et al. [130] applied FT-MIR spectroscopy to discriminate the origin of saffron. Using 250 samples from Greece, Iran, Italy, and Spain, they obtained IR spectra of crushed stigmas through DRIFT and saffron volatile extracts with a ZnSe optical window. Smoothing, baseline correction, and second derivative pretreatments were applied to all IR spectra. Authors performed multivariate PCA-DA analysis in the IR spectra of the crushed stigmas testing different spectral regions, achieving a low recognition ability (67.6%). The best discrimination results were reached in the spectral region 2000–700 cm^−1^ with 98.4% total explained variance and 93.6% classification. Discrimination between Italian samples was attributed to the carbonyl group region (1746 cm^−1^), while the bands at 1600 and 1670 cm^−1^ were responsible for the differentiation of samples from the other countries. Hence, they concluded that FT-MIR spectroscopy along with multivariate analyses is a fast and environmentally friendly method to verify the geographical origin of saffron, which could be applied for quality control, adulteration, and traceability in the international market.

**Table 3 foods-11-00579-t003:** Applications of FT-MIR spectroscopy for saffron quality control.

Spectral Range (cm^−1^)	Sampling Technique	Algorithm	Purpose of the Analysis	Reference
Ground saffron stigmas				
1028 1175–1157	KBr pellets	PCA MLR	Evaluation of the effects of storage conditions and spoilage detection.	[131]
4000–400	Diamond ATR	PCA	Discrimination between pure and adulterated samples (safflower, calendula, and turmeric).	[132]
4000–600 2000–600	DRIFT	PLS-DA PLS	Detection, identification, and quantification of adulteration with saffron stamens, calendula, safflower, turmeric, buddleja, and gardenia.	[133]
1800–1400 1300–700	KBr pellets	PCA PLS-DA	Classification of pure and adulterated samples with carminic acid.	[134]
4000–400	KBr pellets	PCA PLS-DA PLS	Classification by origin, detection, and quantification of adulteration with *C. sativus* style, calendula, safflower, and *Rubia* genus.	[135]
4000–400	Diamond ATR	EPO-PCA EPO-SVM	Classification by origin, detection, and quantification of adulteration with *C. sativus* style, calendula, safflower, and *Rubia* genus.	[136]
Ground stigmas and volatile extracts from Saffron				
2000–700	DRIFT ZnSe window	PCA DA	Classification by geographical origin (Greece, Iran, Italy, and Spain).	[130]
Ground stigmas and aqueous extracts from Saffron				
4000–400	Diamond ATR	SO-PLS-LDA SO-CovSel-LDA	Classification by geographical origin (four zones of Italy).	[137]

PCA: principal component analysis; DA: discriminant analysis; MLR: multiple linear regression; PLS: partial least squares regression; PLS-DA: partial least squares discriminant analysis; SO-PLS-LDA: sequential and orthogonalized partial least squares linear discriminant analysis; SO-CovSel-LDA: sequential and orthogonalized covariance selection linear discriminant analysis; EPO-PCA: external parameter orthogonalization with principal component analysis; EPO-SVM: external parameter orthogonalization combined with support vector machine.

Biancolillo et al. [137], analyzed 114 saffron samples (83 for calibration and 31 for validation) from four Italian areas (Spoleto, L’Aquila, Sicily, and Città della Pieve) via FT-MIR and UV-vis spectroscopy. FT-MIR spectra were obtained from the pulverized saffron samples using a diamond ATR accessory and UV-vis spectra from the aqueous extracts of saffron. They tested diverse combinations of spectral pretreatments (first and second derivative, standard normal variate (SNV), SNV + first derivative, and SNV + second derivative) and simultaneously processed the data from MIR and UV-vis spectra employing two multiblock strategies: sequential and orthogonalized partial least squares linear discriminant analysis (SO-PLS-LDA) and sequential and orthogonalized covariance selection linear discriminant analysis (SO-CovSel-LDA). Better results were obtained with the SO-PLS-LDA model, which only validated 3 of 31 samples erroneously while SO-CovSel-LDA verified 4 of 31 incorrectly. These coincided and were identified as atypical samples from a broader collection area (Sicily), whose variability made their validation difficult.

Ordoudi et al. [131] applied FT-MIR spectroscopy to measure apocarotenoid changes (crocin and picrocrocin) of saffron during storage. They obtained FT-MIR spectra of 52 saffron samples stored in the dark for different periods and analyzed their extracts by UV-vis spectroscopy and HPLC-DAD. FT-MIR spectra were pretreated with smoothing, baseline correction, normalization, and second derivative. Results showed modifications in the 1028 cm^−1^ band associated with glucose residues and in the 1175–1157 cm^−1^ region related to the cleavage of glycosidic bonds. Scores computed in the subsequent PCA analysis were correlated with the data from HPLC-DAD, concluding that the FT-MIR technique constitutes a promising, sensitive and fast tool for saffron quality control. The same authors used FT-MIR spectroscopy to detect carminic acid (CA) in saffron samples [134]. Spectra were pretreated with smoothing, baseline correction, and normalization; then used 1800–1400 and 1300–700 cm^−1^ spectral regions were to perform PCA and PLS-DA. They were able to separate pure samples from those with high concentrations of CA (>10.0%, *w*/*w*) without prior sample preparation except grinding.

Furthermore, Petrakis and Polissiou [133] implemented FT-MIR spectroscopy, DRIFT, and multivariate analyses (PLS-DA and PLS) to identify and quantify plant adulterants in 230 saffron samples: 50 of pure saffron and 30 mixtures of each type of adulterant, i.e., *C. sativus* stamens, calendula, safflower, turmeric, buddleja, and gardenia. Testing different pretreatments (smoothing, baseline correction, and normalization) and performing PLS-DA analysis on data from the fingerprint region of the MIR spectrum (4000–600 cm^−1^), they achieved 99% classification between pure and adulterated saffron (5–20% *w*/*w* levels). They also characterized the type of adulterant employing a six-class PLS-DA model on data from 2000 to 600 cm^−1^ spectral region and quantified each adulterant through PLS methods with detection limits of 1.0–3.1% (*w*/*w*).

Varliklioz et al. [132] compared three spectroscopic techniques (ATR-FTIR, Raman, and LIBS) to classify pure saffron and adulterated samples with safflower, calendula, or turmeric. PCA analysis showed that the ATR-FTIR technique achieved better discrimination results between pure saffron and plant adulterants compared with Raman; nevertheless, the best inter-class classification results were obtained with the laser-induced breakdown spectroscopy (LIBS). Therefore, PLS was performed only with LIBS data.

Recently, Amirvaresi et al. [135] compared NIR and MIR spectroscopic techniques along with multivariate analyses for the detection of adulterants (i.e., *C. sativus* styles, safflower, calendula, and Rubia genus) in saffron samples. The authors examined the spectra of unadulterated samples obtained from each spectroscopic technique by classifying them with PCA according to their origin (warm and cold climate); they obtained a better prediction with NIR spectra. Subsequently, they performed a PLS-DA to discriminate pure samples from the adulterated ones, obtaining satisfactory results with both NIR and MIR spectroscopy. Lastly, they applied a PLS algorithm to quantify adulteration where the MIR spectra did not correlate with the level of adulteration; hence, the concentration of adulterants could not be measured. On a second examination, scholars performed the same classification with FT-MIR spectroscopy (diamond ATR) employing external parameter orthogonalization (EPO) for spectral data pretreatment combined with PCA to classify saffron samples based on their origin (warm and cold climate), obtaining a 90% accurate prediction [136]. Then, they computed an EPO-SVM algorithm to detect adulterated saffron samples, achieving better classification results (>95%) in contrast to other multivariate analyses (RF, QDA, k-NN, PLS-DA, and classification tree). These studies demonstrate the potential of FT-MIR spectroscopy and multivariate analyses for saffron authentication (Table 3).

### 4.4. Vanilla

Vanilla has gained importance in the food industry, is considered native to Mexico, and its uses date back to the first settlers of the country. The *Vanilla* genus belongs to the Orchidaceae family and comprises more than 110 species, of which 15 are aromatic, and only three are cultivated for commercial purposes [138]. Due to vanilla’s extensive usage and high price (after saffron), it is important to determine its quality, yet few studies have implemented FT-MIR spectroscopy on this product (Table 4). Moreno-Ley et al. [139] identified and quantified adulterations in vanilla extracts with ethyl vanillin and coumarin. The model was built with 40 samples adulterated with coumarin (0.25–10 ppm) and 40 samples with ethyl vanillin (0.25–10%). A soft independent modeling of class analogy (SIMCA) was developed to identify and classify the purity of adulterated samples, and PLS1, PLS2, and PCR algorithms to predict the adulterants concentration, resulting in an accurate calibrated model (R^2^ ≥ 0.99). PLS1 algorithm achieved the best prediction performance (R^2^ ≥ 0.99). Another study adequately discriminated between adulterated and unadulterated vanilla samples using SIMCA, PLS-DA, and SVM-C algorithms but could not identify the geographic origin of the samples [140]. Conversely, Sharp et al. [141] used FT-MIR spectroscopy and selected ion flow tube mass spectrometry (SIFT-MS) to formulate an effective model to discriminate the geographical area of vanilla pods according to the composition of the extracts.

### 4.5. Turmeric

Yellow turmeric (*Curcuma longa*) is an ancient spice from South Asia, highly produced in India. Cultivation and demand have increased at a global scale due to its multiple uses; besides seasoning, several studies reported that it has health benefits, e.g., antibacterial, anti-inflammatory, anticancer, and antioxidant properties [142,143,144,145]. The large-scale commercialization and export make it susceptible to adulteration with lower-quality products; hence, IR spectroscopy and multivariate analyses are non-destructive and reliable tools to determine its quality (Table 5).

Dhakal et al. [146] proposed an FT-MIR model to identify the adulteration of yellow turmeric (*Curcuma longa*) with white turmeric (*Curcuma zedoaria*) and Sudan red G dye. A total of 50 yellow turmeric-white turmeric samples were prepared at concentrations of 10%, 20%, 30%, 40%, and 50%: in addition, 50 yellow turmeric-Sudan red G samples at concentrations of 1%, 5%, 10%, 15%, 20%, and 25%. Data were collected in the MIR spectra region and further processed using a PLS regression. PLS regression model was able to estimate adulteration with both Sudan red G dye (R^2^ = 0.97 and RMSEP = 1.3%) and white turmeric (R^2^ = 0.95 and RMSEP = 3.0%) [146].

Yeung et al. [151] conducted a bibliometric study of 18,036 articles published on the biological activity of the main active compound of turmeric: curcumin (diferuloylmethane), and its derivatives, also called curcuminoids. Due to the medicinal interest of these compounds (which can also determine the quality of turmeric) several methods have been proposed to identify them. For instance, Wulandari et al. [147] implemented FT-MIR spectroscopy along with PLS to predict the curcuminoid content in turmeric from different regions of Java Island in Indonesia. Spectra were collected in 4000–650 cm^−1^ spectral region. The predicted concentration values were compared with the actual measures obtained by HPLC, resulting in a high correlation coefficient >0.98. Siregar et al. [149] measured curcumin and desmethoxycurcumin in a pharmaceutical formulation. FT-MIR spectra were subjected to several optimizations, including wavenumber selection and derivatization, to obtain the best prediction models (through PLS regression) for the relationship between actual curcuminoid values determined by HPLC and values calculated by FT-MIR. The coefficient of determination for calibration and validation in the two compounds was >0.99, which indicated an acceptable accuracy of the method [149].

Another application of FTIR analysis was proposed by Rohaeti et al. [150], who applied FT-MIR spectrometry to differentiate between yellow turmeric (*Curcuma longa*) and Java turmeric (*Curcuma xanthorrhiza*) from different regions of Indonesia. Samples of 35 dried and 35 powdered turmeric species were taken. FT-MIR spectra were recorded in the region of 4000–400 cm^−1^. PCA and CVA were computed for species discrimination using 2000–400 cm^−1^ spectral region data, achieving suitable accuracy in species discrimination. On the other hand, Gad and Bouzabata [148] tested an FT-MIR model coupled with PCA and HCA algorithms to classify 30 Curcuma samples by origin (Egypt and Algeria). Curcuminoid content (curcumin, desmethoxycurcumin, and bisdemethoxycurcumin) was previously quantified by HPLC, and then PCR and HCA algorithms were then applied to estimate variations. However, the FT-MIR model could not discriminate between the Egyptian and Algerian samples. The authors attribute the results to both species containing the same curcuminoids but in different concentrations.

### 4.6. Other Spices

Recent studies have successfully applied FT-MIR spectroscopy on other spices (black pepper, paprika, oregano, garlic, onion, and star anise) to detect adulterants (Table 6). For instance, Lohumi et al. [152] used the 1800–0650 cm^−1^ spectral region to detect adulteration in paprika with Sudan I dye, performing a hybrid linear analysis to develop the discrimination model for pure and adulterated samples. McGoverin et al. [153] quantified the adulteration of ground black pepper with buckwheat and millet at concentrations of 5–95% (*w*/*w*) through a PLS algorithm, using spectral regions of 3050–2800 and 1770–550 cm^−1^. In addition, they compared results from the FT-MIR and NIR models. Other authors implemented multivariate analysis models based on FT-MIR and NIR spectra to assess adulteration in black pepper, garlic, onion, and star anise [30,33,154,155].

## 5. Conclusions

The commercialization of coffee, cocoa, and spices faces a severe problem; due to adulteration, it is increasingly difficult to find pure products. This situation affects food companies’ revenues and consumers’ health, making it imperative to implement fast and simple techniques for quality control, for instance, IR spectroscopy. The present review summarized the applications of FT-MIR spectroscopy in conjunction with multivariate analyses for quality control of coffee, cocoa, and spices of high economic value such as saffron, vanilla, turmeric, among others.

FT-MIR spectroscopy and multivariate analysis models have been successfully developed to discriminate between pure and adulterated products, with the ability to quantify the level of adulteration, classify diverse spices according to geographical origin or variety, and measure the content of a specific bioactive compound. Nevertheless, the effectiveness of each model depends on several factors such as the number and variability of samples, sampling technique, spectral regions employed, the analytical technique for compounds quantification, spectral pretreatments, and multivariate analysis algorithms. Increasing the spectra sample size may improve the model’s reliability.

The previously reviewed studies showed that FT-MIR spectroscopy is a reliable, fast, simple, non-destructive, and environmentally friendly technique for quality control, authentication, and traceability of coffee, cocoa, and spices. However, it has been barely implemented in cocoa and spices of high economic value, for example, vanilla, oregano, garlic, onion, and star anise. For spices such as cardamom, cinnamon, cloves, cumin, and ginger, there were no reports of the use of FT-MIR spectroscopy for quality control. This represents a great area of opportunity for further research studies. Another area of opportunity for research suggested is to highlight the detection limits of the method because although FT-MIR spectroscopy has demonstrated detection capacity up to ppb for adulteration of other types of food, it would be important to know the detection limit in the adulteration of spices since in the research reviewed in this work few authors report such values, this would help to establish the field of application of the method.

## Figures and Tables

**Table 2 foods-11-00579-t002:** Applications of FT-MIR spectroscopy on quality control of cocoa.

Spectral Range (cm^−1^)	Sampling Technique	Algorithm	Purpose of the Analysis	Reference
Chocolate				
3600–2800 1800–500	ATR cell	PCA PLS	Determination of cocoa solids content in chocolates.	[113]
1800–700	Diamond ATR	PLS	Quantification and prediction of antioxidant capacity and catechin concentration in chocolate.	[114]
Chocolate and fermented cocoa beans				
4400–600	ZnSe ATR	PLS	Prediction of antioxidant capacity and total phenolic content.	[115]
Cocoa bean shells				
4000–500	Ge ATR	PCA PLS-DA	Identification of systematic patterns related to the geographical origin of the samples.	[116]

PCA: principal component analysis; PLS: partial least squares regression; SIMCA: soft independent modeling of class analogy; PLS-DA: partial least squares discriminant analysis.

**Table 4 foods-11-00579-t004:** Applications of FT-MIR spectroscopy on quality control of vanilla.

Spectral Range (cm^−1^)	Sampling Technique	Algorithm	Purpose of the Analysis	Reference
Ethanolic vanilla extracts				
4000–700	ZnSe ATR	SIMCA	Determination of origin according to the main compounds of the vanilla pods.	[141]
3000–1100 1700–1110 1800–850	ZnSe ATR	PLS1 PLS2 PCR	Quantification of adulteration with ethyl vanillin and coumarin.	[139]
4000–700 1549–778	ZnSe ATR	PCA; SIMCA PLS-DA SVM-C	Discrimination between pure and adulterated samples as well as by origin (from Madagascar or other than Madagascar).	[140]

PCR: principal component regression; PCA: principal component analysis; SIMCA: soft independent modeling of class analogy; PLS1: partial least squares with single y-variables; PLS2: partial least squares with multiple y-variables; PLS-DA: partial least squares discriminant analysis; SVM-C: support vector machine-classification mode.

**Table 5 foods-11-00579-t005:** Applications of FT-MIR spectroscopy in quality control of turmeric.

Spectral Range (cm^−1^)	Sampling Technique	Algorithm	Purpose of the Analysis	Reference
Yellow turmeric powder				
1700–700	Ge ATR	PLS	Adulteration with Sudan red G dye.	[146]
1820–1172	Ge ATR	PLS	Prediction of total and individual curcuminoid composition.	[147]
4000–400	KBr pellets	PCA HCA	Discrimination between turmeric from Egypt and Algeria.	[148]
White turmeric powder				
1700–900	Ge ATR	PLS	Adulteration with Sudan red G dye.	[146]
Curcuminoid tablets				
2975–660 1784–1587	Diamond ATR	PLS PCR	Quantification of curcuminoids (curcumin and desmethoxycurcumin).	[149]
Ethanolic extract of *Curcuma longa* and *Curcuma xanthorrhiza*				
4000–400 2000–400	KBr pellets	PCA CVA	Discrimination and identification between *Curcuma longa* and *Curcuma xanthorrhiza.*	[150]

PCR: principal component regression; PLS: partial least squares regression; HCA: hierarchical cluster analysis; PCA: principal component analysis; CVA: canonical variate analysis.

**Table 6 foods-11-00579-t006:** Applications of FT-MIR spectroscopy in quality control of other spices of commercial interest.

Spectral Range (cm^−1^)	Sampling Technique	Algorithm	Purpose of the Analysis	Reference
Black pepper				
3050–2800 1770–550	Diamond ATR	PLS	Comparison of NIR and MIR techniques to quantify the level of adulteration with buckwheat and millet in black pepper.	[153]
4000–400	DRIFT	PCA GA-SVM PLS-DA	Classification of pure pepper and pepper adulterated with sorghum or Sichuan pepper (5–50%).	[156]
3800–2800 1800–400	Diamond ATR	PCA OPLS-DA	Comparison between NIR and MIR to detect adulteration of black pepper with peels, pinheads, spent material, papaya, and chili seeds.	[155]
4000–720	Ge ATR	PCA	Application of microscopy and FT-MIR spectroscopy to detect organic and mineral adulterants in black pepper.	[157]
Paprika				
1800–650	Diamond ATR	HLA	Detection of paprika adulteration with Sudan I dye.	[152]
3300–2700 1800–400	Diamond ATR	PCA SIMCA	Detection of paprika adulteration with adulterants (Sudan I and IV, lead chromate, lead oxide, among others).	[158]
4000–400	Diamond ATR	SO-PLS-LDA SO-CovSel-LDA	Authentication of Senise bell pepper and detection of adulteration with ordinary paprika.	[159]
Oregano				
3999–2800 1800–550	Diamond ATR	PCA OPLS-DA	Detection of oregano adulteration with olive, hazelnut, myrtle, cistus g, and sumac leaves.	[160]
4000–600	Diamond ATR	PCA PLS	FT-MIR detection of adulteration in oregano and quantification by LC-MS/MS.	[161]
Garlic powder				
4000–650 1500–650 1666–1508	Diamond ATR	PLS PLS-SR PLS-VIP	Prediction of adulteration of garlic powder with cornstarch (1–35% *w*/*w*).	[162]
4000–550	Diamond ATR	PCA OPLS-DA	Comparison of NIR and MIR for the detection of different adulterants in garlic.	[30]
Onion powder				
4000–650	Diamond ATR	PCA PLS	Quantification of onion adulteration with cornstarch (1–35% *w*/*w*) using NIR and MIR spectroscopy.	[33]
Star anise powder				
4000–400	KBr pellets	PCA LDA	Comparison of NIR and MIR spectroscopy and the combination of both techniques to detect adulteration of star anise with lower-quality species.	[154]

PLS: partial least squares regression; PCA: principal component analysis; GA-SVM: genetic algorithm optimized support vector machine; PLS-DA: partial least squares discriminant analysis; OPLS-DA: orthogonal partial least square discriminant analysis; HLA: hybrid linear analysis; SIMCA: soft independent modeling of class analogy; SO-PLS-LDA: sequential and orthogonalized partial least squares linear discriminant analysis; SO-CovSel-LDA: sequential and orthogonalized covariance selection linear discriminant analysis; PLS-SR: partial least squares regression with selectivity ratios; PLS-VIP: partial least squares regression with variable importance in projection; LDA: linear discriminant analysis.

## Data Availability

Not applicable.

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
