# Peer review of "Uses of FT-MIR Spectroscopy and Multivariate Analysis in Quality Control of Coffee, Cocoa, and Commercially Important Spices"

_foods, 2022, doi:10.3390/foods11040579_

Round 1

Reviewer 1 Report

The review is interesting and well structured. All the main applications are described and the references appear complete and robust. I just have some considerations and a few minor revisions.

Main comments:

  • Sine the review is mainly centred on the application of multivariate techniques, a brief theorethical part explaining the most important differences among the several technqies that were reviewed shliud be provided; the same is true for pre-treatment methods, since the spectra pre-treatment deeply affects the final result.
  • The reviwe reports many cases of applications to adulteration studies but usually these studies are applied to identify models able to discriminate between adulterated and not adulterated samples. I think it is imporant to add a comment in the conclusions about the importance of carrying out studies about the identification of the lowest level of adulteration that can be identified in a certain sample, to state the field of applicability of the method.

Minor revisions are given below:

  • Line 56: change “species” with “spices”
  • Line 77: TOF is not defined
  • Line 85: change”in different spices” with “to different spices”
  • Line 106: The MIR spectrum
  • Line 111-112: please reformulate the sentence since it is not clear
  • Line 119: the obtained spectra
  • Line 120: CaF2
  • Line 125-126: i do not fully agree with the highest reproducibility of ATR sectroscopy since it deeply depends on the pressure it is applied to the sample
  • Line 148: it seems that too many blank spaces are present after “namely”
  • Line 200: FT-MIR spectroscopy and ….
  • Line 224: in validation
  • Line 233: quantified, with
  • Line 234: technique, the above mentioned adulterants
  • line 235: and a percentage
  • line 237: the data
  • line 244: The FT-MIR model presented a lower
  • line 281: autoscaling
  • line 363: Afterwards,
  • line 366: Trinitario and Forastero
  • line 369: it has better yields.
  • Line 378: of variety.
  • Line 378: the works on FT-MIR spectroscopy application to cocoa and chocolate.
  • Line 411: 14% of tested chocolates differed
  • Line 413: change performed with built
  • Line 415: it is not clear the meaning of “acceptable in predicting its values”, please provide numbers
  • Line 418: accuracy 0.74-0.96: why values so different?
  • Line 423: naturally reproduced
  • Line 436: the red colouring
  • Line 439: and an anticarcinogen
  • Line 497: UV-vis spectroscopy
  • Line 527: spectroscopic
  • Line 530: from the adulterated ones
  • Line 572-587: it refers to turmeric
  • Line 656: change robustness with reliability
  • Line 657: The previously reviewed studies

Author Response

Dear Editor,

The authors have reviewed the reviewers' comments and made the suggested changes, which have been marked in green to Reviewer 1 and yellow to Reviewer 2.

The changes made to each observation are detailed below.

Reviewer 1:

Major Comments:

Sine the review is mainly centred on the application of multivariate techniques, a brief theorethical part explaining the most important differences among the several technqies that were reviewed shliud be provided; the same is true for pre-treatment methods, since the spectra pre-treatment deeply affects the final result.

The reviwe reports many cases of applications to adulteration studies but usually these studies are applied to identify models able to discriminate between adulterated and not adulterated samples. I think it is imporant to add a comment in the conclusions about the importance of carrying out studies about the identification of the lowest level of adulteration that can be identified in a certain sample, to state the field of applicability of the method.

Authors reply

Thank you for your valuable comments; I briefly included a theoretical part on the main multivariate analyses used (PCA and PLS). Además se incluyó la descripción de los pretratamientos espectrales más utilizados (Normalization, Baseline correction, Smoothing and Derivative).

I added a coment in the conclusions about the importance of establishing the detection limit in the adulteration of a given sample to be applied in future studies.

Minor Revisions:

  • Line 56: change “species” with “spices”

Authors reply

Thank you for the observation; I changed “species” with “spices”.

  • Line 77: TOF is not defined

Authors reply

Thanks for the comment; the meaning of TOF was added.

  • Line 85: change ”in different spices” with “to different spices”

Authors reply

Thank you for the observation; I changed “in different spices” with “to different spices”.

  • Line 106: The MIR spectrum

Authors reply

Thanks for the comment; I added "The" to "MIR spectrum".

  • Line 111-112: please reformulate the sentence since it is not clear

Authors reply

Thanks for the comment; I rephrased the sentence.

  • Line 119: the obtained spectra

Authors reply

Thanks for the comment; I changed “the spectra obtained” to “the obtained spectra”.

  • Line 120: CaF2

Authors reply

Thank you for the observation; I changedCaF2” with “CaF2

  • Line 125-126: i do not fully agree with the highest reproducibility of ATR sectroscopy since it deeply depends on the pressure it is applied to the sample

Authors reply

Thanks for the comment; however, with today's equipment, it is easily reproducible since they have pressure control to adjust the samples to the same pressure.

  • Line 148: it seems that too many blank spaces are present after “namely”

Authors reply

Thank you for the observation; I performed the correction.

  • Line 200: FT-MIR spectroscopy and ….

Authors reply

Thank you for the observation; I added the missing word.

  • Line 224: in validation

Authors reply

Thank you for the observation; I changed “on” with “in”.

  • Line 233: quantified, with

Authors reply

Thanks for the comment; I added the comma.

  • Line 234: technique, the above mentioned adulterants

Authors reply

Thank you for the observation; I have made the correction mentioned above.

  • line 235: and a percentage

Authors reply

Thank you for the observation; I changed “and percentage” with “and a percentage”.

  • line 237: the data

Authors reply

Thank you for the observation; I changed “data” with “the data”.

  • line 244: The FT-MIR model presented a lower

Authors reply

Thank you for the observation; I have made the correction mentioned above.

  • line 281: autoscaling

Authors reply

Thank you for the observation; I changed “autoscale” with “autoscaling”

  • line 363: Afterwards,

Authors reply

Thanks for the comment; I added the letter “s”.

  • line 366: Trinitario and Forastero

Authors reply

  • Thanks for the comment; I changed “y” with “and”.
  • line 369: it has better yields.

Authors reply

Thank you for the observation; I deleted the sentence “in the amount of fruits”.

  • Line 378: of variety.

Authors reply

Thank you for the observation; I deleted the sentence “it would be”.

  • Line 378: the works on FT-MIR spectroscopy application to cocoa and chocolate.

Authors reply

Thank you for the observation; I have made the correction mentioned above.

  • Line 411: 14% of tested chocolates differed

Authors reply

Thanks for the comment; I changed “14% of chocolates tested differed” with “14% of tested chocolates differed”.

  • Line 413: change performed with built

Authors reply

Thanks for the comment; I changed “performed” with “built”.

  • Line 415: it is not clear the meaning of “acceptable in predicting its values”, please provide numbers

Authors reply

Thank you for the observation; the coefficients of determination obtained in the prediction of the aforementioned work were added.

  • Line 418: accuracy 0.74-0.96: why values so different?

Authors reply

Thanks for the comment; I have placed the correct accuracy value obtained by the authors.

  • Line 423: naturally reproduced

Authors reply

Thanks for the comment; I changed “reproduce naturally” with “naturally reproduced”.

  • Line 436: the red colouring

Authors reply

Thanks for the comment; I changed “red coloration” with “the red colouring”.

  • Line 439: and an anticarcinogen

Authors reply

Thank you for the observation; I added the word "an"

  • Line 497: UV-vis spectroscopy

Authors reply

Thank you for the observation; I added the word "spectroscopy".

  • Line 527: spectroscopic

Authors reply

Thanks for the comment; I changed “spectrometry” with “spectroscopic”.

  • Line 530: from the adulterated ones

Authors reply

Thanks for the comment; I changed “from those adulterated” with “from the adulterated ones”

  • Line 572-587: it refers to turmeric

Authors reply

Excuse me for the error; I have placed these paragraphs in the corresponding section (turmeric).

  • Line 656: change robustness with reliability

Authors reply

Thanks for the comment; I changed “robustness” with “reliability”.

  • Line 657: The previously reviewed studies

Authors reply

Thank you for the observation; I changed “The studies previously reviewed” with “The previously reviewed studies”.

Reviewer 2 Report

The current review artcle deals with the authentication of coffee, cocoa, and commercially important spices using mainly FT-IR analysis and chemometrics. The topic is of interest for researchers and food sectors. The authors have in general prepared well their review and the body of the text can be easily followed. I have, however, some indications for authors to improve their study. My comments are in the attached pdf file. Based on these comments, I suggest a minor revision prior to further consideration .

Author Response

Dear Editor,

The authors have reviewed the reviewers' comments and made the suggested changes, which have been marked in green to Reviewer 1 and yellow to Reviewer 2.

The changes made to each observation are detailed below.

Reviewer 2:

The current review article deals with the authentication of coffee, cocoa, and commercially important spices using mainly FT-IR analysis and chemometrics. The topic is of interest for researchers and food sectors. The authors have in general prepared well their review and the body of the text can be easily followed. I have, however, some indications for authors to improve their study. My comments are in the attached pdf file. Based on these comments, I suggest a minor revision prior to further consideration.

  • Line 18: change they with these

Authors reply

Thanks for the comment; I changed “they” with “these”.

  • Line 26: were utilized

Authors reply

Thank you for the observation; I added the word "were".

  • Line 29: change they with these

Authors reply

Thanks for the comment; I changed “They” with “These”.

  • Line 58: change they with these

Authors reply

Thanks for the comment; I changed “they” with “these”.

Line 66: no italic is required

Authors reply

Thanks for the observation; however, this word does not appear in italics in the document.

  • Line 362: change they with these

Authors reply

Thanks for the comment; I changed “they” with “these”.

  • Line 421: The relevant study ''European Food Research and Technology, 243(9), 1577-1591 (2017)'' which is into the topic has not been included. It presents additional work from Greece, Spain, Iran. etc.

Authors reply

I appreciate your valuable suggestion; however, in this study they use multivariate analysis but don´t use mid-infrared spectroscopy for this reason it was not included in the review.

  • Table 3: ground. Correct through the text.

Authors reply

Thanks for the observation; I made the change from "Grounded" to "Ground" through the text.
